# Efficacy and Safety of Checkpoint Inhibitor Treatment in Patients with Advanced Renal or Urothelial Cell Carcinoma and Concomitant Chronic Kidney Disease: A Retrospective Cohort Study

**DOI:** 10.3390/cancers13071623

**Published:** 2021-04-01

**Authors:** Florian Seydel, Susanne Delecluse, Martin Zeier, Tim Holland-Letz, Georg Martin Haag, Anne Katrin Berger, Barbara Christine Grün, Nina Bougatf, Markus Hohenfellner, Stefan Duensing, Dirk Jäger, Stefanie Zschäbitz

**Affiliations:** 1Department of Nephrology, University Hospital Heidelberg, 69120 Heidelberg, Germany; florian.seydel@web.de (F.S.); s.delecluse@dkfz.de (S.D.); martin.Zeier@med.uni-heidelberg.de (M.Z.); 2German Center for Infection Research (DZIF), 69120 Heidelberg, Germany; 3German Cancer Research Centre (DKFZ) Unit F100, 69120 Heidelberg, Germany; 4German Cancer Research Centre (DKFZ) Unit C060, 69120 Heidelberg, Germany; t.holland-letz@dkfz.de; 5Department of Medical Oncology, National Center of Tumor Diseases (NCT), University Hospital Heidelberg, 69120 Heidelberg, Germany; GeorgMartin.Haag@med.uni-heidelberg.de (G.M.H.); anne.berger@med.uni-heidelberg.de (A.K.B.); barbara.gruen@med.uni-heidelberg.de (B.C.G.); dirk.jaeger@nct-heidelberg.de (D.J.); 6Cancer Registry, National Center of Tumor Diseases (NCT), University Hospital Heidelberg, 69120 Heidelberg, Germany; nina.bougatf@med.uni-heidelberg.de; 7Department of Urology, University Hospital Heidelberg, 69120 Heidelberg, Germany; markus.hohenfellner@med.uni-heidelberg.de (M.H.); stefan.duensing@med.uni-heidelberg.de (S.D.)

**Keywords:** checkpoint inhibitor, chronic kidney disease, renal cell carcinoma, urothelial carcinoma

## Abstract

**Simple Summary:**

Immune checkpoint inhibition plays a pivotal role in the treatment of metastatic renal cell carcinoma and metastatic urothelial carcinoma. The association of chronic kidney disease with these tumors is well established. However, to what extent kidney failure modifies the efficacy or the toxicity profiles of checkpoint inhibitors has been poorly investigated. In this paper, we reviewed the files of 85 patients with renal cell carcinoma and 41 with urothelial cancer who had received checkpoint inhibitor treatment, and found that 37.6% and 41.5% had evidence of chronic kidney disease, respectively. We found that neither general treatment-related nor immune-related adverse events differed between patients with normal or impaired renal function. Using a multivariate analysis, we found that chronic kidney disease had no effect on progression-free survival. However, irrespective of the tumor entity, chronic kidney disease was found to positively influence overall survival. We conclude that treatment with checkpoint inhibitors in patients with chronic kidney disease is safe and efficient.

**Abstract:**

*Background*: Checkpoint inhibitors are a standard of care in the treatment of advanced renal cell carcinoma (RCC) and urothelial carcinoma (UC). Patients with these tumors often suffer from concomitant chronic kidney disease (CKD). Limited data are available on the efficacy and toxicity of checkpoint inhibitors in patients with CKD. *Methods*: We retrospectively analyzed 126 patients who received checkpoint inhibitors for RCC (*n* = 85) or UC (*n* = 41) and analyzed the frequency of treatment- and immune-related adverse events (AEs). We performed a multivariate analysis to determine progression-free survival (PFS) and overall survival (OS). *Results*: A total of 38.9% of patients had CKD. Frequencies of general AEs (49.0% in CKD vs. 48.1%, *p* > 0.99999) and immune-related AEs (28.6 vs. 24.7%, *p* ≥ 0.9999) did not significantly differ between the groups. There was no difference in PFS for patients with RCC or UC and CKD or without CKD (RCC: 6.81 vs. 7.54 months, HR 1.000 (95%CI 0.548–01.822), *p* = 0.999; UC:2.33 vs. 3.67 months, HR 01.492 (95%CI 0.686–3.247), *p* = 0.431). CKD appeared to be a potential effect modifier for OS in both RCC and UC (RCC: NR vs. 23.9 months, HR 0.502 (95%CI 0.219–1.152), *p* = 0.104; UC:18.84 vs. 15.42 months, HR 0.656 (95%CI 0.296–1.454), *p* = 0.299). *Conclusions*: Checkpoint inhibitor treatment in our cohort of patients with CKD was as safe and efficient as in the cohort of patients without CKD.

## 1. Introduction

The use of checkpoint inhibitors (CPI) has become a standard of care (SOC) for patients with locally advanced/metastatic renal cell carcinoma (RCC) and urothelial carcinoma (UC). Large randomized trials have demonstrated the efficacy of CPI in patients with advanced RCC. Four first-line combination therapies have been approved: nivolumab in combination with ipilimumab [1], axitinib plus pembrolizumab [2], axitinib plus avelumab [3], and cabozantinib plus nivolumab [4]. In addition, nivolumab is approved for previously treated patients [5]. In UC, the sequence of therapeutic agents is dependent upon fitness for cisplatin/carboplatin. Either a maintenance approach with avelumab directly following platinum-based induction chemotherapy [6] or CPI treatment upon progression is included in the therapeutic regimen [7,8,9,10,11]. 

CPI have well-known toxicity profiles. In a meta-analysis the frequencies of any grade adverse events (AE) and grade 3–4 AEs have been reported as 66.0% and 14%, respectively [8]. Immune-related AE (irAE) are a direct consequence of the CPI mechanism of action. Endocrine, dermatologic, and gastrointestinal irAEs are among the most common. The risk of an irAE occurrence is dependent of the agent, dose, and type of regimen (monotherapy vs. combination therapy), but is also higher in patients with preexisting autoimmune disease [9].

Chronic kidney disease (CKD) is characterized by a loss of kidney function and a reduction of glomerular filtration rate (GFR). CKD is connected to RCC and UC in multiple ways. As a consequence of surgical tumor treatment, the 3-year probability of developing CKD rises, depending on the surgical method, up to 20–65% [10]. Moreover, it is widely accepted that CKD increases the risk of tumor development. Tumors of the urinary tract, and in particular RCC, have an up to 10 times higher incidence rate in patients with CKD [11]. The molecular mechanisms that underlie this increased risk remain undefined.

CPI lead to the reactivation of exhausted anti-tumor T cells that are part of the tumor microenvironment. Previous findings suggested that uremia causes a functional impairment of CD4+ T cells and accounts for a weak response after vaccination against hepatitis B [12]. In vitro experiments in uremic milieu showed a decrease in T cell activation and T cell proliferation, which dampens the adaptive immune response [13]. Limited information on CPI efficacy and toxicity in patients with concomitant kidney failure is available. AEs were reported for a single CPI agent, atezolizumab, in patients with UC, but no differences in efficacy and safety between patients with or without CKD were found [14,15]. Here, we report progression-free survival (PFS), overall survival (OS), and frequency of AE and irAE in patients with CKD who received multiple types of CPI, both for RCC or UC at the National Center of Tumor Diseases (Heidelberg, Germany). Thus, our study extends previously published work, both in the type of CPI used and in the tumor types in which they were given.

## 2. Methods

### 2.1. Study Design

Data from 129 patients who received treatment with CPI for locally advanced or metastatic RCC or UC (mRCC, mUC) at the National Center of Tumor Diseases (University Hospital of Heidelberg, Heidelberg, Germany), during the years 2015 to 2019 were retrospectively collected. CKD was determined by a reduction of the estimated GFR (eGFR) below 60 mL/min/1.73 m^2^ over a 3-month period. Patients with end-stage renal disease were excluded from our analysis (*n* = 3). CKD staging was performed as suggested by the Kidney Disease Improving Global Outcome (KDIGO) guidelines [16]. Patients with CKD were compared to patients with normal renal function (eGFR >60 mL/min/1.73 m^2^); the latter were referred to as non-CKD patients. The study was approved by the Ethics Committee of the University of Heidelberg (S-034-2020).

### 2.2. Assessments

OS was defined as the time interval between the inception of CPI therapy and death. PFS was defined as the time from inception of CPI therapy to documented disease progression. Disease assessments were performed with computed tomography or magnetic resonance imaging, according to local standard and current treatment guidelines for RCC and UC [17,18]. In case of tumor progression or treatment discontinuation, patients were followed up for the occurrence of AE and survival. AEs of all grades were included in our analysis and graded according to the National Cancer Institute Common Terminology Criteria for Adverse Events, version 5.0 [19]. AEs were systematically recorded in an electronic patient chart during routine patient care. Data were analyzed retrospectively. 

### 2.3. Statistics

Data were summarized using descriptive statistics and reported as either arithmetic means with standard deviation (SD), hazard ratio (HR), median with 95% confidence interval (CI), or percentages. Differences for categorical variables were analyzed using Fischer’s tests, multivariate testing for survival data was performed using Cox regression analysis, and the results were presented as HRs (95% CIs). The Cox model included patients’ age at CPI initiation, gender, location of metastasis and line of treatment. All statistical analyses were performed using the GraphPad Prism 8 (Graphpad Software, San Diego, CA, USA), SigmaPlot 13.0 (Systat Software, San Jose, CA, USA) and R version 3.6.1 (R Foundation, r-project.org, last accessed 29 March 2021) software packages.

## 3. Results

### 3.1. Patient Collective

A total of 126 patients were treated with CPI from 2015 to 2019. Of these, 85 patients had mRCC and 41 had mUC. We identified 49 (38.9%) patients with CKD and eGFR below 60ml/min/m^2^. Of those, 17 had mUC (17/41, 41.5%) and 32 had mRCC (32/85, 37.6%). Characteristics of patients with and without CKD (gender, type of carcinoma (RCC vs. UC), synchronous or metachronous metastatic status, site of distant metastasis, histological grading, and International Metastatic Renal Cell Carcinoma Database (IMDC) risk group (only in case of RCC) are displayed in Table 1). Patients with CKD were significantly older than those without (68.52 ± 10.21 years vs. 61.39 ± 11.36 years, *p* = 0.0005). 

### 3.2. Exposure and Safety 

In total, 84 patients (67.7%) received treatment with nivolumab, 22 (17.5%) with nivolumab plus ipilimumab, 10 (7.9%) with pembrolizumab, eight (6.3%) with atezolizumab, one (0.8%) with pembrolizumab plus axitinib, and one (0.8%) with tremelimumab plus durvalumab (Table 3).

**Table 3 cancers-13-01623-t003:** Characteristics of checkpoint inhibitor therapy. CKD = chronic kidney disease; CPI = Checkpoint inhibitor; RCC = renal cell carcinoma; UC = urothelial carcinoma; ns = not significant; * = significant; ** = very significant.

Characteristics of CPI Therapy	CKD *n* = 49	Non-CKD *n* = 77	*p*
Type of CPI—*n* (%)			
Pembrolizumab	7 (14.3)	3 (3.9)	* (0.0460)
Pembrolizumab + axitinib	0 (0.00)	1 (1.3)	ns (>0.9999)
Nivolumab	32 (65.3)	52 (67.5)	ns (0.84759)
Atezolizumab	4 (8.2)	4 (5.2)	ns (0.2067)
Nivolumab + ipilimumab	6 (12.2)	16 (20.8)	ns (0.1459)
Tremelimumab + durvalumab	0 (0.00)	1 (1.3)	ns (>0.9999)
Number of doses—median ± SD	15.00 ± 20.49	12.95 ± 12.43	ns (0.7089)
Line of treatment (RCC)	N = 32 (100.0)	N = 53 (100.0)	
1st	7 (21.9)	14 (26.4)	ns (0.7963)
2nd	19 (59.4)	27 (50.9)	ns (0.5053)
3rd	3 (9.4)	7 (13.2)	ns (0.7362)
4th	2 (6.3)	3 (5.7)	ns (>0.9999)
5th	1 (3.1)	2 (3.8)	ns (>0.9999)
Prior treatment (RCC) (1st–4th line)	N = 35 (100.0)	N = 58 (100.0)	
Sunitinib	13 (37.1)	34 (58.6)	ns (0.0555)
Pazopanib	14 (40.0)	8 (13.8)	** (0.0056)
Cabozantinib	4 (11.4)	9 (15.5)	ns (0.7603)
Interferon	1 (2.9)	0 (0.0)	ns (0.3763)
Axitinib	1 (2.9)	2 (3.4)	ns (>0.9999)
Temsirolimus	1 (2.9)	0 (0.0)	ns (0.3763)
Everolimus	1 (2.9)	3 (5.2)	ns (>0.9999)
Sorafenib	0 (0.0)	1 (1.7)	ns (>0.9999)
Bevacizumab + interferon	0 (0.0)	1 (1.7)	ns (>0.9999)
Line of treatment (UC) (1st–4th line)	N = 17 (100.0)	N = 24 (100.0)	
1st	6 (35.3)	2 (8.3)	* (0.0486)
2nd	10 (58.8)	14 (58.3)	ns (>0.9999)
3rd	1 (5.9)	7 (29.2)	ns (0.1102)
4th	0 (0.0)	1 (4.2)	ns (>0.9999)
Prior treatment (UC)	N = 11 (100.0)	N = 22 (100.0)	
Cisplatin-backbone	6 (54.5)	18 (81.8)	ns (0.1210)
Carboplatin-backbone	3 (27.3)	3 (13.6)	ns (0.3752)
Gemcitabine mono	2 (18.2)	1 (4.5)	ns (>0.9999)

CPI were most often applied as a second-line treatment (*n* = 70, 56.5%). Prior treatment in patients with RCC or UC and CKD was not different from that given to patients without CKD and included mainly sunitinib and panzopanib (RCC) and Cisplatin (UC) (Table 3). Furthermore, 54.4% of mUC patients with CKD received a cisplatin-based chemotherapy prior to CPI therapy. Treatment-related AE of any grade occurred at similar rates in patients with or without CKD (49.0 vs. 48.1%, *p* > 0.99999) (Table 4). However, patients without CKD suffered significantly more often from diarrhea than patients with CKD (15.6 vs. 2.0%, *p* = 0.0157).

**Table 4 cancers-13-01623-t004:** Adverse events (AE) during checkpoint inhibitor therapy. RA = rheumatoid arthritis. CKD = chronic kidney disease; ns = not significant; * = significant.

Adverse Events—*n* (%)	CKD *n* = 49	Non-CKD *n* = 77	*p*
Patients with adverse event	24 (49.0)	37 (48.1)	ns (>0.99999)
All adverse events	35	81	
Type of adverse event			
Diarrhea	1 (2.0)	12 (15.6)	* (0.0157)
Vomiting	5 (10.2)	7 (9.1)	ns (>0.99999)
Constipation	5 (10.2)	5 (6.5)	ns (0.5093)
Nausea	5 (10.2)	9 (11.7)	ns (>0.99999)
Loss of appetite	2 (4.1)	8 (10.4)	ns (0.3137)
Fatigue	4 (8.2)	13 (16.9)	ns (0.1909)
Rash	5 (10.2)	14 (18.2)	ns (0.3086)
Pruritus	2 (4.1)	6 (7.8)	ns (0.4819)
Infusion reaction	0 (0.0)	2 (2.6)	ns (0.5209)
Mucositis	1 (2.0)	1 (1.3)	ns (>0.99999)
Exacerbation RA	1 (2.0)	0 (0.0)	ns (>0.99999)
Xerostomia	0 (0.0)	3 (3.9)	ns (0.2813)
Acute kidney injury	4 (8.2)	1 (1.3)	ns (0.0746)

We observed irAEs of any grade in 14/49 (28.6%) and 19/77 (24.7%) of patients with and without CKD, respectively (*p* ≥ 0.9999). One grade-5 irAE (pneumonitis) occurred in a patient with CKD (Table 5).

**Table 5 cancers-13-01623-t005:** Immune-related adverse events during checkpoint inhibitor therapy. CTCAE = common toxicity criteria for adverse events; elev. = elevation; irAE = immune-related adverse event; pancr. = pancreatic; insuff. = insufficiency. CKD = chronic kidney disease; ns = not significant.

irAE—no. (%)	CKD *n* = 49	Non-CKD *n* = 77	*p*
Patients with irAE	14 (28.6)	19 (24.7)	ns (>0.9999)
All irAE	15	23	
Type of irAE			
Colitis	1 (2.0)	6 (7.8)	ns (0.2459)
Nephritis	4 (8.2)	1 (1.3)	ns (0.0746)
Elev. of liver enzymes	1 (2.0)	3 (3.9)	ns (>0.99999)
Dermatitis	2 (4.1)	2 (2.6)	ns (0.6419)
Elev. of pancr. enzymes	0 (0.0)	3 (3.9)	ns (0.2813)
Gastritis	0 (0.0)	2 (2.6)	ns (0.5209)
Pneumonitis	4 (8.2)	1 (1.3)	ns (0.0746)
Myocarditis/myositis	0 (0.0)	1 (1.3)	ns (>0.99999)
Arthritis/myalgia	0 (0.0)	1 (1.3)	ns (>0.99999)
Thyroid dysfunction	2 (4.1)	2 (2.6)	ns (0.6419)
Adrenocortical insuff.	0 (0.0)	1 (1.3)	ns (>0.99999)
Encephalitis	1 (2.0)	0 (0.0)	ns (>0.99999)
CTCAE-stage of irAE			
1	3 (6.1)	2 (2.6)	ns (0.3762)
2	9 (18.4)	10 (13.0)	ns (0.4503)
3	1 (2.0)	9 (11.7)	ns (0.0871)
4	1 (2.0)	2 (2.6)	ns (>0.99999)
5	1 (2.0)	0 (0.0)	ns (>0.99999)
Therapy of irAE			
Glucocorticoids	11 (22.4)	21 (27.3)	ns (0.6753)
Therapy interruption	7 (14.3)	8 (10.4)	ns (0.5775)
Therapy discontinuation	4 (8.2)	11 (14.3)	ns (0.4018)
Therapy continued	4 (8.2)	4 (5.2)	ns (0.7102)

Nephritis and pneumonitis were the most common irAEs in patients with CKD (*n* = 4, 8.2%), whereas colitis was most common in patients without CKD (*n* = 6; 7.8%). There was no statistical difference regarding involved organ or grade of irAE. Treatment of irAEs included corticoids, other immunosuppressive therapy, interruption, or discontinuation of CPI. Treatment-related irAEs leading to treatment interruption or discontinuation were similar between patients with and patients without CKD (*p* = 0.5775 and *p* = 0.4018) (Table 5).

### 3.3. Efficacy

The median duration of treatment in patients with CKD and RCC was 5.39 months (95%CI 2.80-11.60) and similar to that of patients with normal eGFR (5.20 months; 95%CI 3.73–7.60; *p* = 0.4520) (Table 6). 

In total, two patients (3.8%) without CKD achieved a complete response (CR), and seven (21.9%) and 10 (18.9%) of the patients with and without CKD, respectively, had a partial response (PR). Most patients, however, had progressive disease (PD) (CKD: *n* = 16 (50%) vs. non-CKD: *n* = 22 (41.51%)) or SD (CKD: *n* = 9 (28.1%) vs. non-CKD: *n* = 19 (35.8%)). There were no statistical differences in the response category or number of ongoing responses between the two groups. The median time to response (4.22 and 3.10 months) and duration of therapy response (13.37 and 7.47 months) were not statistically different in patients with or without CKD (Table 6). During the median follow-up time of 15.49 (95%CI: 6.70–24.23) and 11.43 (95%CI:6.63–16.57) months, PFS was 6.81 months for CKD patients with RCC (95%CI:3.98–35.42) and 7.54 months for patients without renal impairment (95%CI:4.76-NR) (Figure 1A). 

CKD appeared to be a potential effect modifier for overall survival in RCC that, however, did not reach statistical significance (NR, 95%CI:15.27-NR vs. 23.9 months, 95%CI:18.19-NR; HR 0.502 (95%CI 0.219–1.152), *p* = 0.104) (Figure 1B and Table 7). 

For patients with UC, median duration of treatment was 2.33 months (95%CI:1.37–11.97) for patients with CKD and 4.88 (95%CI:1.80–35.67) months for patients without CKD (*p* = 0.3110). No CR was documented. PR as best response was documented in four (23.5%) patients with and four (16.7%) patients without CKD. In the majority of patients, PD was documented as best response to CPI treatment (CKD: *n* = 11 (64.7%) and non-CKD: *n* = 13 (54.2%)) (Table 6).

We found no difference in the median PFS for patients with CKD (2.30 months; 95%CI:1.36–11.75) or without CKD in the UC cohort (4.81 months; 95%CI:1.73–23.76, HR 1.492 (95%CI:0.686–3.247; *p* = 0.431)) (Figure 1C). The median OS was 18.84 (95%CI:2.38-NR) in CKD patients with UC versus 15.42 months (95%CI:7.11–50.33) in patients with UC without CKD (Figure 1D and Table 7). CKD appeared to mildly modify OS (18.84 vs. 15.42 months, HR 0.656 (95%CI:0.296–1.454), *p* = 0.299). However, the mean follow-up time of patients with UC and CKD (8.87 months 95%CI:2.30–17.63) was shorter than the mean follow-up of patients without CKD (14.64; 95%CI:3.42–22.60) (Table 6). 

## 4. Discussion

Immune checkpoint inhibition plays a pivotal role in the treatment of mRCC and mUC. The association of CKD with RCC and UC is well established [10,11]. To what extent CKD modifies efficacy or toxicity profiles of CPI has been poorly investigated [20]. We reviewed the files of 126 patients with RCC or UC who received CPI treatment at our institution and found that 32 (37.6%) patients with RCC and 17 (41.5%) patients with UC had evidence of CKD (eGFR < 60mL/min/m^2^). In 75.5% of the cases, CKD was a consequence of tumor surgery. This is in line with previous reports that found up to 26% of the patients with RCC have CKD prior to surgery and that this number rises to 65% after radical nephrectomy [10]. The patients investigated in our study were treated over similar time periods and received a similar number of CPI doses, irrespective of their kidney function. During the period of observation, approximately half of the patients in each group (CKD and non-CKD) developed AE of any grade (49.0% vs. 48.1%, ns) and, except for diarrhea, there was no difference in the organs involved. Immune-related AEs of any grade were noted in 12 and 18 patients, 24.5% and 23.4%, respectively (*p* = 0.8863). The safety profile of CPI in both groups was comparable for patients with and without CKD and seemed more favorable than in previous reports, where general side effects and treatment-related AEs of any degree were found in 60–98% and 46–79% of patients treated with single agent CPI [5,6,21,22,23,24,25]. In a prospective cohort of patients treated with atezolizumab for UC safety was not significantly different between patients with normal and impaired renal function (any grade AE: 88% (all) vs. 80% (CKD); treatment-related AE: 53% (all) vs. 39% (CKD)) [14]. Furthermore, in a cohort of 214 patients that received atezolizumab and of which 102 patients had CKD, treatment-related AE rate was comparable between CKD and non-CKD patients and found in 38–54% [15]. Our data were prospectively recorded as part of routine patient care. Prior to every CPI application, a physician visit followed by a specialist oncology nurse visit were performed. Therefore, we are confident that all clinically relevant side effects have been recorded.

Skin, GI tract, and endocrine glands are the organs most affected by irAEs during RCC and UC treatment [26,27,28,29]. Interestingly, colitis was the most common irAE in our patients without CKD, followed by hepatic and endocrine irAE, whereas in patients with CKD, nephritis/elevated creatinine was the most common irAE, followed by colitis and dermatologic irAEs. Previous studies showed that acute kidney injury potentially caused by PD-1 inhibitors pembrolizumab or nivolumab occurs in about 2.2% (95% CI:1.5–3.0) of patients treated with CPI [30]. Furthermore, 3.2–5.4% of patients treated with nivolumab and ipilimumab showed an elevation of creatinine [31]. We found a similar percentage of nephritis in our patients’ collective (4.0%). While in the non-CKD group only one patient (1.3%) was affected, elevation of creatinine/nephritis was noted in four patients (8.2%) of the CKD-group. Although not formally proven by biopsy, the typical clinical presentation, together with an absence of other pre- and post-renal causes, response to steroids and reversibility of symptoms with normalization of GFR deem it highly likely that nephritis caused the creatinine elevation. A possible explanation for our findings is that nephritis is a common irAE that becomes clinically visible only in patients with underlying kidney insufficiency. 

To our knowledge, this is the first study that investigates the efficacy of various CPI in patients with RCC and UC. We found no differences in PFS for RCC and UC patients with and without CKD, although well-known confounders such as age, gender, and location of metastasis were considered in our analysis. However, we identified CKD as an effect modifier for OS that nearly reached statistical significance. It is worth noting that approximately 20% of the patients in our cohort received CPI in 3rd to 5th line and therefore in higher treatment lines than in the pivotal trials. The median OS was not reached in CKD RCC patients and was 23.64 months in patients without CKD. For patients with UC, OS was 18.84 months in CKD patients, and was 15.47 months in patients with normal renal function. The survival of the latter group was longer than reported in previous real-world datasets and prospective trials [6,14,21,25,32]. Our finding is surprising at first sight, as CKD has often been associated with worse OS, potentially as a result of the ineligibility of these patients for adjuvant and/or standard treatments [33]. However, in our cohort, a high proportion of UC CKD patients, 54.4%, still received cisplatin-based chemotherapy. Preclinical data support a synergism between cisplatin and PD-L1 blockade through enhanced T cell responses and PD-L1 expression on tumor cells [34,35]. Larger cohorts of patients with CKD and CPI treatment are needed to confirm this result. Potential limitations of our study include its retrospective character and its moderate sample size.

## 5. Conclusions

We conclude from our findings that CPI treatment in patients with CKD and metastatic RCC or UC can be safely administered and is an efficient treatment option.

## Figures and Tables

**Figure 1 cancers-13-01623-f001:**
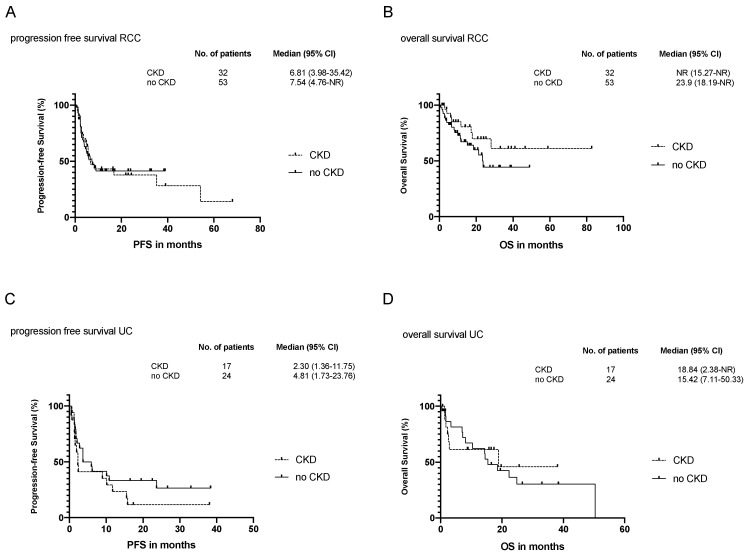
(**A**) Progression-free survival (PFS) of patients with RCC and CKD or without CKD (non-CKD). The percentage of patients without tumor progression is plotted as unadjusted PFS. Median PFS and 95% confidence intervals (95%CI) are given in months. Hazard ratio (HR) with 95%CI and statistical differences between CKD and non-CKD group as result of a multivariate Cox regression analysis are indicated. (**B**) Overall survival (OS) of patients with RCC and CKD or without CKD (non-CKD). The percentage of patients alive is plotted as unadjusted OS. Median OS and 95% confidence intervals (95%CI) are given in months. HR with 95%CI and statistical differences between CKD and non-CKD group as result of a multivariate Cox regression analysis are indicated. (**C**) Same as in (**A**) but for patients with UC. (**D**) Same as in (**B**) but for patients with UC.

**Table 1 cancers-13-01623-t001:** Patients’ characteristics. CKD = chronic kidney disease; IMDC = International Metastatic Renal Cell Carcinoma Database; RCC = renal cell carcinoma; SD = standard deviation; UC = urothelial carcinoma; ns = not significant; * = significant; *** = extremely significant.

Characteristics	CKD *n* = 49	Non-CKD *n* = 77	*p*
Median age—years ± SD	68.52 ± 10.21	61.39 ± 11.36	*** (0.0005)
<65 years—no. (%)	18 (36.7)	47 (61.0)	* (0.0104)
≥65 years—no. (%)	31 (63.3)	30 (39.0)	* (0.0104)
Sex—*n* (%)			
Male	36 (73.5)	63 (81.8)	ns (0.2752)
Female	13 (26.5)	14 (18.2)	ns (0.2752)
Tumor entity—*n* (%)			
UC	17 (34.7)	24 (31.2)	ns (0.7004)
RCC	32 (65.3)	53 (68.8)	ns (0.7004)
IMDC-Score—*n* (%) (RCC only)	*n* = 32	*n* = 53	
Favorable	5 (15.6)	14 (26.4)	ns (0.2927)
Intermediate	18 (56.3)	29 (54.7)	ns (>0.9999)
Poor	9 (28.1)	10 (18.9)	ns (0.4213)
Grading RCC—*n* (%)			
G1	0 (0.00)	1 (1.89)	ns (>0.9999)
G2	16 (50.00)	27 (50.94)	ns (>0.9999)
G3	10 (31.25)	11 (20.75)	ns (0.3076)
G4	1 (3.13)	2 (3.77)	ns (>0.9999)
Unknown	5 (15.63)	12 (22.64)	ns (0.5782)
Grading UC—*n* (%)			
G1	0 (0.00)	1 (4.17)	ns (>0.9999)
G2	5 (29.41)	3 (12.5)	ns (0.2412)
G3	9 (52.94)	12 (50.0)	ns (>0.9999)
G4	0 (0.00)	0 (0.00)	ns (>0.9999)
Unknown	3 (17.64)	8 (33.33)	ns (0.3092)
Metastatic status—*n* (%)			
Synchronous	19 (38.8)	25 (32.5)	ns (0.5658)
Metachronous	30 (61.2)	52 (67.5)	ns (0.5658)
Sites of distant metastasis—*n* (%)			
Lung	36 (73.5)	52 (67.5)	ns (0.5528)
Lymph nodes	25 (51.0)	46 (59.8)	ns (0.3617)
Bone	21 (42.9)	35 (45.5)	ns (0.8548)
Liver	19 (38.8)	25 (32.5)	ns (0.5658)
Adrenal gland	6 (12.2)	11 (14.3)	ns (0.7960)
Brain	3 (6.1)	14 (18.2)	ns (0.0638)
Pleura	6 (12.2)	10 (13.0)	ns (>0.9999)
Peritoneum	7 (14.3)	7 (9.1)	ns (0.3946)
Skin	1 (2.0)	0 (0.00)	ns (0.3889)
Other	17 (34.7)	27 (35.1)	ns (>0.9999)

CKD was in most cases attributed to prior nephrectomy (75.5%) or nephrotoxic tumor therapy (22.5%). Diabetes mellitus and hypertension were underlying diseases in 9 (18.4%) patients with CKD (Table 2).

**Table 2 cancers-13-01623-t002:** Causes for chronic kidney disease.

Causes of CKD—*n* (%)	CKD *n* = 49
Nephrectomy	37 (75.5)
Nephrotoxic chemotherapy	11 (22.5)
Diabetes mellitus	8 (16.3)
Hypertensive nephropathy	1 (2.0)
Hydronephrosis	1 (2.0)
Primary (atherosclerotic) cirrhotic kidney	1 (2.0)

**Table 6 cancers-13-01623-t006:** Efficacy of checkpoint inhibitor therapy. CKD = chronic kidney disease; RCC = renal cell carcinoma; UC = urothelial carcinoma; ns = not significant.

Efficacy of CPI Therapy	RCC CKD *n* = 32	RCC Non-CKD *n* = 53	*p*	UC CKD *n* = 17	UC Non-CKD *n* = 24	*p*
**Best overall response**—*n* (%)						
Complete response	0 (0.0)	2 (3.8)	ns (0.5249)	0 (0.0)	0 (0.0)	ns (>0.99999)
Partial response	7 (21.9)	10 (18.9)	ns (0.7837)	4 (23.5)	4 (16.7)	ns (0.6975)
Stable disease	9 (28.1)	19 (35.8)	ns (0.4873)	2 (11.8)	7 (29.2)	ns (0.2623)
Progressive disease	16 (50.0)	22 (41.5)	ns (0.5035)	11 (64.7)	13 (54.2)	ns (0.5387)
**Median time** **to response**—months (95%CI)	4.22 (3.00–51.73)	3.10 (2.7–3.77)	ns (0.1003)	3.33 (2.30–5.63)	3.95 (1.83–7.30)	ns (0.6709)
**Median duration** **of response**—months (95%CI)	13.37 (3.17–35.8)	7.47 (3.03–12.27)	ns (0.0537)	9.82 (6.23–14.10)	18.97 (1.07–35.67)	ns (0.3717)
**Patients with** **ongoing response**—*n* (%)	4 (12.50)	9 (16.98)	ns (0.7584)	1 (5.88)	2 (8.33)	ns (>0.99999)
**Median duration** **of therapy**—months (95%CI)	5.39 (2.8–11.60)	5.20 (3.73–7.60)	ns (0.2347)	2.33 (1.37–11.87)	4.88 (1.80–19.44)	ns (0.3110)
**Median follow-up**—months (95%CI)	15.49 (6.70–24.23)	11.43 (6.63–16.57)	ns (0.0770)	8.87 (2.30–17.63)	14.64 (3.43–22.60)	ns (0.2416)

**Table 7 cancers-13-01623-t007:** Results of a multivariate Cox regression analysis of patients with and without CKD. CKD = chronic kidney disease; RCC = renal cell carcinoma; UC = urothelial carcinoma, * = vs. first line treatment.

	HR	CI 95%	*p*
**Progression-Free Survival RCC**			
CKD	1.000	0.548–1.822	0.999
Sex	1.044	0.473–2.300	0.916
Age	1.010	0.982–1.040	0.481
Distant metastasis bone and or brain and or liver	1.409	0.641–2.676	0.334
CPI treatment 2nd line *	1.121	0.529–2.378	0.765
CPI treatement 3–5th line *	1.423	0.593–3.415	0.430
**Progression-Free Survival UC**			
CKD	1.492	0.686–3.247	0.431
Sex	0.983	0.381–2.534	0.733
Age	1.021	0.981–1.063	0.311
Distant metastasis bone and or brain and or liver	1.141	0.553–2.353	0.623
CPI treatment 2nd line *	1.901	0.698–5.180	0.209
CPI treatement 3–5th line *	1.556	0.441–5.490	0.492
**Overall Survival RCC**			
CKD	0.502	0.219–1.152	0.104
Sex	0.840	0.319–2.209	0.724
Age	1.039	1.002–1.077	0.039
Distant metastasis bone and or brain and or liver	1.229	0.463–3.265	0.679
CPI treatment 2nd line *	1.025	0.366–1.025	0.963
CPI treatement 3–5th line *	1.582	0.496–5.041	0.438
**Overall Survival UC**			
CKD	0.656	0.296–1.454	0.299
Sex	1.737	0.774–3.899	0.181
Age	0.741	0.310–1.774	0.502
Distant metastasis bone and or brain and or liver	0.522	0.138–1.971	0.338
CPI treatment 2nd line *	2.447	1.020–5.874	0.045
CPI treatement 3–5th line *	0.997	0.985–1.009	0.599

## Data Availability

All original data and materials are available upon request from the corresponding author.

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
