# Peer review of "Efficacy and Safety of Checkpoint Inhibitor Treatment in Patients with Advanced Renal or Urothelial Cell Carcinoma and Concomitant Chronic Kidney Disease: A Retrospective Cohort Study"

_cancers, 2021, doi:10.3390/cancers13071623_

Round 1

Reviewer 1 Report

This is a retrospective study investigating the association between chronic kidney disease and clinical outcomes and toxicity of check-point inhibitors for patients with either renal or urothelial carcinoma. Despite the topic being of interest in clinical practice, my major concern is that the retrospective nature of the analysis, the small population (RCC=85 and UC=41), the spurity of the stage of disease (some were metastatic, some locally advanced), and the various checkpoint inhibitors most likely bias the results and certainly prevent from drawing definitive conclusions. Conversely, both in the abstract and the manuscript state that checkpoint inhibitors are safe and active for patients with RCC or UC. In addition, I have the following comments: 

Simple Summary - This section has many acronyms which per journal guidelines should be avoided. Also, CPI should be fully spelled both in Simple Summary and Abstract.

Abstract - The conclusion does not reflect the outcomes of this small retrospective study as the results cannot be generalized. Also, the last sentence is ambiguous "We conclude that CPI treatment in patients with CKD is safe and efficient" as authors do not specify what patients with CKD they are referring to.

Introduction - This section is unnecessarily lenghty and should be shortened. 

Methods - It would appear that data from 126 patients were collected from this study but "Patients with end-stage renal disease were excluded from our analysis (n=3)." However, in Results authors state that the total population is still 126 patients. 

Results -

a) Authors state that patients are "well balanced regarding ....site of distant metastases". However, table 1 shows clear differences in rate of lymphnodes, brain, liver, and peritoneal metastases between the 2 cohorts.   

b) "CKD appeared to be a relevant effect modifier for overall survival in RCC that nearly reached statistical significance". This sentence has no scientific basis. How can CKD relevantly modify OS when the P is not even significant.

Discussion - This section is very speculative and authors tend to make cases of very small numbers within cohorts. 

Author Response

Comment 1-1: Simple Summary - This section has many acronyms which per journal guidelines should be avoided. Also, CPI should be fully spelled both in Simple Summary and Abstract.

Response to comment 1-1: We modified our short summary which is now free of acronyms and we have fully spelled CPI in both the short summary and abstract.

Comment 1-2: Abstract - The conclusion does not reflect the outcomes of this small retrospective study as the results cannot be generalized. Also, the last sentence is ambiguous "We conclude that CPI treatment in patients with CKD is safe and efficient" as authors do not specify what patients with CKD they are referring to.

Response to comment 1-2: We have relativized our conclusion and limited it to our cohort of patients. Moreover, we have made clear that we refer to efficacy and safety in patients without renal failure. 

Comment 1-3: Introduction - This section is unnecessarily lengthy and should be shortened. 

Response to comment 1-3: We have shortened the introduction (page 5, lines 69-78).

Comment 1-4: Methods - It would appear that data from 126 patients were collected from this study but "Patients with end-stage renal disease were excluded from our analysis (n=3)." However, in Results authors state that the total population is still 126 patients. 

Response to comment 1-4: We thank the reviewer for this comment and have corrected the numbers to 129 (total) and 126 (excluding patients with dialysis) (page 7, line 106).

Comment 1-5: Results - Authors state that patients are "well balanced regarding ....site of distant metastases". However, table 1 shows clear differences in rate of lymph nodes, brain, liver, and peritoneal metastases between the 2 cohorts.

Response to comment 1-5: We agree that the percentages of patients with distant metastasis are not numerically equal between the two groups. However, differences between groups were not statistically different. We changed the paragraph to: ”Characteristics of patients with and without CKD (gender, type of carcinoma (RCC vs. UC), synchronous or metachronous metastatic status, site of distant metastasis, histological grading and IMDC risk group in case of RCC) are displayed in table 1. Patients with CKD were significantly older than those without (68.52 +10.21 years vs. 61.39+11.36 years, p=0.0005).” on page 9 lines 140-143.

Comment 1-6: Results -  "CKD appeared to be a relevant effect modifier for overall survival in RCC that nearly reached statistical significance". This sentence has no scientific basis. How can CKD relevantly modify OS when the P is not even significant.

Response to comment 1-6: We respectfully disagree with the reviewer’s opinion on this point. We performed a multivariate analysis on OS that showed that survival (in months, CKD vs. non-CKD) in RCC was NR vs. 23.9 months, HR 0.508 (95%CI 0. 222-1.165), p=0.110 and in UC was 18.84 vs. 15.42 months, 0.737 (95%CI 0.343-1.581), p=0.433), p=0.070. Statistical significance was just missed in both cases. However, hazard ratios in both cases implied that patients with CKD died at lower rate than the control population without CKD. Given the data, this is the best possible estimate for the true CKD effect. From a statistical perspective, it would be wrong to deny an assertion on the basis of a just missed p value (Altmann et al., Statistics notes: Absence of evidence is not evidence of absence, BMJ, 1995). However, we modified our statement to “potential effect modifier that, however, did not reach statistical significance”.

Comment 1-7: Discussion - This section is very speculative, and authors tend to make cases of very small numbers within cohorts. 

Response to comment 1-6: We thank the reviewer for his comment. We have removed speculative statements about mechanisms that might be involved in better CPI response in CKD patients. However, we already stated in the first version of the manuscript that larger studies are needed to confirm our results.

Reviewer 2 Report

It was my pleasure reviewing "Efficacy and safety of checkpoint inhibitor treatment in patients with advanced renal or urothelial cell carcinoma and concomitant chronic kidney disease: a retrospective cohort study" by Seydel et al. The manuscript presents a retrospective analysis of 86 renal and 41 urothelial cases. Of these 49 had CKD. On multivariable analysis CKD was not associated with PFS but was associated with OS "nearly reached statistical significance". The manuscript has many issues. 1. The study has two different tumor types. I do not understand why authors felt renal which is a slow growing tumor can be combined with urothelial which is more aggressive. 2. There is treatment heterogeneity with patients receiving PD-1, PD-1+VEGF TKI, PD-1+CTLA-4. How can they be combined together for analysis. 3. PD-1 treatment was administered at different lines 4. The authors state they did multivariable analysis but do not provide data in that regard. Again the sample size is too small and heterogeneous to account for all variables. 5. There is no term like "nearly reached statistical significance". Either something is statistically significant or it is not. 6. Finally and most importantly it is well known that checkpoint inhibitors are safe in CKD. There have been pharmacokinetic studies in this regard. This manuscript is not bringing anything new for scientific community.

Author Response

Comment 2-1: The study has two different tumor types. I do not understand why authors felt renal which is a slow growing tumor can be combined with urothelial which is more aggressive.

Response to comment 2-1: The aim of our study was to understand whether chronic kidney failure influences the efficacy of CPI therapy. We have chosen two tumor entities that are frequently treated with CPI. We agree that UC and RCC are different tumor types that display different growth characteristics. To this end we performed two separate multivariate analyses for efficacy, one for UC and one for RCC. With respect to side effects, we feel confident that analyzing patients irrespective of their tumor type is reasonable since the general side effects of CPI have been shown to be equally distributed between patients with different tumor types(Xu C, Chen YP, Du XJ, Liu JQ, Huang CL, Chen L, et al. Comparative safety of immune checkpoint inhibitors in cancer: systematic review and network meta-analysis. BMJ. 2018;363:k4226).

Comment 2-2: There is treatment heterogeneity with patients receiving PD-1, PD-1+VEGF TKI, PD-1+CTLA-4. How can they be combined together for analysis.

Response to comment 2-2:

We agree with this reviewer that there is treatment heterogeneity within our cohort. With the exception of one patient that received PD-1-inhibitor in combination with TKI all patients received either single PD-L1/ PD-1-antibody or the combination of PD-L1/ PD-1 antibody with CTLA-4 antibody.We performed statistical analyses to compare differences in CPI regimens between patients with and without CKD. With the exception of pembrolizumab, we could not find any statistically significant difference between different agents. Since the combination of PD-L1/ PD-1 antibody with CTLA-4 is associated with higher numbers of side effects, it might have been sensible to compare general AEs and irAEs in patients with and without CKD who had received a combination of PD-L1 or PD-1 antibody, together with CTLA-4. However, the highly divergent number of patients included in both groups (6 versus 43 and 17 versus 60) precludes any meaningful statistical analysis. Thus, treatment heterogeneity did not have any influence on our main conclusions.

Comment 2-3: PD-1 treatment was administered at different lnes.

Response to comment 2-3: We agree with the reviewer that CPI were given in different treatment lines. However, we could not find any statistical differences between patients with CKD and no CKD and the line of CPI treatment (Tab. 3). Regarding toxicity there is no clear evidence of differences between treatment line and/ or ECOG PS (the reminder being addressed by several trials) (Chauhan A, Kabir T, Wu J, Wei J, Cook M, Kunos CA. Prognostic and predictive factors associated with ipilimumab-related adverse events: a retrospective analysis of 11 NCI-sponsored phase I clinical trials. Oncotarget. 2020;11:1427-34.). Nearly 80% of our patients received CPI as a first or second line. Therefore, we feel confident that there is no bias due to the line of CPI treatment.

Comment 2-3: The authors state they did multivariable analysis but do not provide data in that regard. Again, the sample size is too small and heterogeneous to account for all variables.

Response to comment 2-4. We now provide the results of our multivariate analyses (Tab. 7). We agree that it is impossible to include all possible variables that could have influenced our results. However, we feel confident that we included all necessary and important modifiers of OS und PFS in patients with RCC and UC such as age, gender and site of distant metastasis.

Comment 2-5: There is no term like "nearly reached statistical significance". Either something is statistically significant or it is not.

Response to comment 2-5: We respectfully disagree with the reviewer’s opinion. This item of criticism was mentioned by referee 1, please see our answer 1-6.

Comment 2-6: Finally and most importantly it is well known that checkpoint inhibitors are safe in CKD. There have been pharmacokinetic studies in this regard. This manuscript is not bringing anything new for scientific community.

Response to comment 2-6: We carefully read the literature and could not find any comparable study that analyzed CPI efficacy and safety in patients with CKD. As already stated in the introduction of our manuscript, AEs were reported for a single CPI agent, atezolizumab, in patients with UC, but no differences in efficacy and safety between patients with or without CKD were found. The study of Hoffmann-Censits et al. is limited to response rates to CPI and does not report OS or PFS. In both studies irAEs were not reported. It might be that our results chime with the reviewer’s own clinical experience or that they appear to be intuitive. However, to the best of our knowledge, the topic addressed in our paper is poorly characterized in the literature. Thus, our study significantly extends previously published work, both in the type of CPI used and in the tumor types in which they were given.

Reviewer 3 Report

Seydel et al report the results of a retrospective study conducted in 85 patients with metastatic renal cell carcinoma (RCC) and 41 patients with urothelial cell carcinoma (UC), treated with immune checkpoint inhibitors (ICI). The aim of this study is to assess the toxicity profiles of treatment with immune checkpoint inhibitor in patients with chronic kidney disease (CKD).

The median follow-up was 11.4 to 15.5 months in the groups of patients with RCC, and 2.9 to 7.4 months in patients with UC. Regarding the baseline characteristics, no significant difference was observed between the groups of patients with and without CKD with the exception of age which was slightly higher in patients with CKD.

Overall, immune-related adverse events occurred at similar rates in patients with and without CKD. The overall survival trends to be higher in patients with CKD, although the differences are not significant compared with non-CKD patients. The authors conclude that treatment with checkpoint inhibitors is safe and efficient in patients with CKD.

The study is well conducted and statistical analyses are appropriate. However, some points require clarification and some conclusions regarding the positive impact of CKD on response to ICI may be moderated.

Major comments

Results section

Line 209: “CKD appeared to be a relevant effect modifier for overall survival in RCC”

Although the trend is clear, the difference remains non significant. The authors should moderate this sentence for ex replacing the term “relevant” by the term “potential”.

Have the authors tested stratification by CKD stage or creatinin clearance, as in the article of Hoffman-Censtis JITC 2020, to further assess if this trend is majored in patients with more severe CKD?

Line 222: “Again, CKD appeared as relevant effect modifier for OS that nearly reached statistical significance”.

In the case of UC, the survival curves do not display any trend of higher benefit in patients with CKD; can the authors provide the HR and the p values of the log rank tests?

Line 223: “However, the mean follow-up time of patients with UC and CKD (2.87 month 95%CI:2.10-8.87) was shorter than the mean follow-up of patients without CKD (7.40 95%CI:3.43-18.77) (Tab 6).”

This is indeed a potential limitation; an update of the follow-up would enable to clarify the weight of this difference in the follow-up.

Table 1:

Inclusion of tumor grade would be interesting, since an imbalance of this parameter across the different groups is likely to influence the clinical outcome, especially the OS.

Cox regression:

Why has tumor grade not been included as a potential confounding factor? Results of the Cox regression should be presented in a separate table rather than in Figure 1; only P values and HR of log rank tests should be displayed on this figure.

Table 2:

In 75.5% of cases, nephrectomy is the cause of CKD. Can this parameter be considered as a bias of the study, since non-CKD patients were potentially also non-operable patients?

Discussion section

“Furthermore, one could speculate that CPI substances are more effective in patients with CKD. It is known that patients with CKD have higher levels of inflammatory markers and suffer from systemic inflammation induced by reactive oxygen species (ROS) [36]. Therefore, it is possible that CPI targets are expressed at higher levels in these patients.”

This statement is speculative and may not remain as a potential explanation of the trend in OS improvement in the group of patients with CKD unless additional arguments are provided.

Line 250: “Our data were collected retrospectively, and AEs were not systematically recorded. However, prior to every CPI application, a physician visit followed by a specialist oncology nurse visit were performed. Therefore, we are confident that all clinically relevant side effects have been recorded”

This point is not very clear and does not really matches the previous sentence. Were the AE systematically notified in the frame of the current care and collected retrospectively? If yes, this point may be simplified and integrated to the materials and methods section.

Minor comments

The layout of tables 3 and 6 requires some adjustments

In figure 1, the legends should be modified to avoid “B) Same as in A) but for patients with overall survival (OS) and RCC. C) Same as in A) but for urothelial cancer (UC). D) Same as in B) but for UC.”

The authors may use “NS” or an equivalent instead of the exact p value in the discussion section, the exact p value being already displayed in the tables.

Author Response

Comment 3-1: Line 209: “CKD appeared to be a relevant effect modifier for overall survival in RCC”. Although the trend is clear, the difference remains non significant. The authors should moderate this sentence for ex replacing the term “relevant” by the term “potential”.

Response to comment 3-1: We thank the reviewer for his comment, we have replaced “relevant” by “potential”.

Comment 3-2: Have the authors tested stratification by CKD stage or creatinin clearance, as in the article of Hoffman-Censtis JITC 2020, to further assess if this trend is majored in patients with more severe CKD?

Response to comment 3-2: We thank the reviewer for this comment. We have not separately tested the influence of a severe renal impairment with eGFR<30ml/min/qm since we had only five patients with this level of renal insufficiency in our cohort. Three of them had end stage renal disease and were excluded.

Comment 3-3: Line 222: “Again, CKD appeared as relevant effect modifier for OS that nearly reached statistical significance”.

Response to comment 3-1: We have replaced “relevant” by “potential”.

Comment 3-4: In the case of UC, the survival curves do not display any trend of higher benefit in patients with CKD; can the authors provide the HR and the p values of the log rank tests?

Response to comment 3-4: We now provide the results of our multivariate analyses (Tab. 7).

Comment 3-5: Line 223: “However, the mean follow-up time of patients with UC and CKD (2.87 month 95%CI:2.10-8.87) was shorter than the mean follow-up of patients without CKD (7.40 95%CI:3.43-18.77) (Tab 6).” This is indeed a potential limitation; an update of the follow-up would enable to clarify the weight of this difference in the follow-up.

Response to comment 3-5: We thank the reviewer for his comment on this issue. We have updated the period of follow-up. As a result, follow-up for UC which was 2.5 times longer for patients without CKD is now only 1.5 times longer. We have accordingly updated our multivariate analyses, but observed only slight changes that did not change the main conclusions of the paper and are now included into the results section, Figure 1 and Table 6 and 7.

Comment 3-6: Table 1. Inclusion of tumor grade would be interesting, since an imbalance of this parameter across the different groups is likely to influence the clinical outcome, especially the OS.

Response to comment 3-6: We thank the reviewer for his suggestion and now provide the wanted data separately for UC and RCC in Table 1. We could not find any statistically significant differences between the groups. Therefore, we did not include tumor grading into our regression model.

Comment 3-7: Why has tumor grade not been included as a potential confounding factor?

Answer 3-7. This issue is discussed in answer 3-6.

Comment 3-8: Results of the Cox regression should be presented in a separate table rather than in Figure 1; only P values and HR of log rank tests should be displayed on this figure.

Response to comment 3-8: We now provide the results of our multivariate analyses (Tab. 7).

Comment 3-9: Table 2: In 75.5% of cases, nephrectomy is the cause of CKD. Can this parameter be considered as a bias of the study, since non-CKD patients were potentially also non-operable patients?

Response to comment 3-9: We thank the reviewer for his comment. All but 7 (13.21%) patients without CKD and RCC and 1 (3.13%) patient with CKD and RCC had a complete or partial nephrectomy. Hence, only a minority of patients in both groups were non-operable and nephrectomy has therefore not been considered as potential confounder. 

Comment 3-10: Discussion section. “Furthermore, one could speculate that CPI substances are more effective in patients with CKD. It is known that patients with CKD have higher levels of inflammatory markers and suffer from systemic inflammation induced by reactive oxygen species (ROS) [36]. Therefore, it is possible that CPI targets are expressed at higher levels in these patients.” This statement is speculative and may not remain as a potential explanation of the trend in OS improvement in the group of patients with CKD unless additional arguments are provided.

Response to comment 3-10: We have already addressed this issue in our answer to point 1-7.

Comment 3-11: Line 250: “Our data were collected retrospectively, and AEs were not systematically recorded. However, prior to every CPI application, a physician visit followed by a specialist oncology nurse visit were performed. Therefore, we are confident that all clinically relevant side effects have been recorded”. This point is not very clear and does not really matches the previous sentence. Were the AE systematically notified in the frame of the current care and collected retrospectively? If yes, this point may be simplified and integrated to the materials and methods section.

Response to comment 3-10: We thank the reviewer for his/her comment and have included the requested information into the method section of our manuscript (page 7, lines 124-125). We have modified our statement in the discussion part (page 19, lines 273-274). AEs were systematically recorded during patient care and collected retrospectively.

Comment 3-12: The layout of tables 3 and 6 requires some adjustments

Response to comment 3-12: We have changed the layout of both tables and hope that this has improved readability.

Comment 3-13: In figure 1, the legends should be modified to avoid “B) Same as in A) but for patients with overall survival (OS) and RCC. C) Same as in A) but for urothelial cancer (UC). D) Same as in B) but for UC.”

Response to comment 3-13: We have modified the figure legend and hope that this has improved readability.

Comment 3-14: The authors may use “NS” or an equivalent instead of the exact p value in the discussion section, the exact p value being already displayed in the tables.

Response to comment 3-14: We have modified the discussion part and replaced p values by “ns”.

Round 2

Reviewer 1 Report

Acceptable now.

Author Response

We thank the reviewer for the positive comments.

Reviewer 2 Report

The authors have provided the multivariate model which has multiple flaws. They have treated bone mets (good prognosis) as same as brain or liver (bad prognosis). They have not included lines of therapy as overall survival will be different based on subsequent therapies. The authors need to correct these before multivariable model can be meaningful.

Author Response

We have corrected our multivariate analysis according to the comments of this referee. We have also included lines of treatment as requested (Tab. 7).

Reviewer 3 Report

Figure 1

The authors should provide the HR and the P values of the univariate analysis obtained with log rank tests rather than the results of the Cox regressions, which are already displayed in Table 7.

Minor:

In panels A an B, the survival curve of the CKD group is displayed in dashed line, whereas in panels C and D, the survival curve of the CKD group is displayed in solid line. A homogenous presentation of the survival curves would facilitate the reading of the figure.

Line 246: ”Again, CKD appeared as potential effect modifier for OS (18.84 vs. 15.42 months, HR 246 0.737 (95%CI:0.343-1.581), p=0.433)). “

The increase of the follow-up and the subsequent modification of the p value for OS weaken this conclusion. The authors should consider the suppression of this statement.

Author Response

Statement 3-1:

Figure 1

The authors should provide the HR and the P values of the univariate analysis obtained with log rank tests rather than the results of the Cox regressions, which are already displayed in Table 7.

Answer 3-1. We have deleted the results of the Cox regression analyses from Fig. 1. However, we feel that it is not helpful to include the results of the univariate analysis in this figure, since it does not provide meaningful information.  

Statement 3-2:

Minor:

In panels A an B, the survival curve of the CKD group is displayed in dashed line, whereas in panels C and D, the survival curve of the CKD group is displayed in solid line. A homogenous presentation of the survival curves would facilitate the reading of the figure.

Answer 3-2. We have now used the same line formats in all figures.

Statement 3-3:

Line 246: ”Again, CKD appeared as potential effect modifier for OS (18.84 vs. 15.42 months, HR 246 0.737 (95%CI:0.343-1.581), p=0.433)). “

The increase of the follow-up and the subsequent modification of the p value for OS weaken this conclusion. The authors should consider the suppression of this statement.

Answer 3-3. We have weakened our statement that now reads: “CKD appeared to mildly modify OS (18.84 vs. 15.42 months, HR 246 0.737 (95%CI:0.343-1.581), p=0.433)).”

Round 3

Reviewer 2 Report

I would like to thank the authors for accommodating the suggested changes. I do not have any further questions or suggestions.